# N-Gram Induction Heads for In-Context RL: Improving Stability and Reducing Data Needs

## ABSTRACT

In-context learning allows models like transformers to adapt to new tasks from a few examples without updating their weights, a desirable trait for reinforcement learning (RL). However, existing in-context RL methods, such as Algorithm Distillation (AD), demand large, carefully curated datasets and can be unstable and costly to train due to the transient nature of in-context learning abilities. In this work, we integrated the n-gram induction heads into transformers for in-context RL. By incorporating these n-gram attention patterns, we considerably reduced the amount of data required for generalization and eased the training process by making models less sensitive to hyperparameters. Our approach matches, and in some cases surpasses, the performance of AD in both grid-world and pixel-based environments, suggesting that n-gram induction heads could improve the efficiency of in-context RL.

## 1 INTRODUCTION

In-context learning is a powerful ability of pretrained autoregressive models, such as transformers [30] or state-space models [12]. In contrast to fine-tuning, in-context learning is able to effectively solve downstream tasks on inference without explicitly updating the weight of a model, making it a versatile tool for solving wide range of tasks [1]. Originated in the language domain [4], the in-context ability has quickly found its applications in Reinforcement Learning (RL) for building agents that can adaptively react to the changes in the dynamics of the environment. This trait allows researchers to use In-Context Reinforcement Learning (ICRL) as a backbone for the embodied agents [7] or to benefit from its adaptation abilities for domain recognition in order to build generalist agents [11, 24].

In-context reinforcement learning methods that learn from offline datasets were first introduced by Laskin et al. [17] and Lee et al. [18]. In the former work, Algorithm Distillation (AD), authors propose to distill the policy improvement operator from a collection of learning histories of RL algorithms. After pretraining a transformer on these learning histories, an agent is able to generalize to unseen tasks entirely in-context. In the latter approach, the authors show it is possible to train adaptive models on datasets that contain interactions collected with expert policies, provided that the optimal actions for each state are also available.

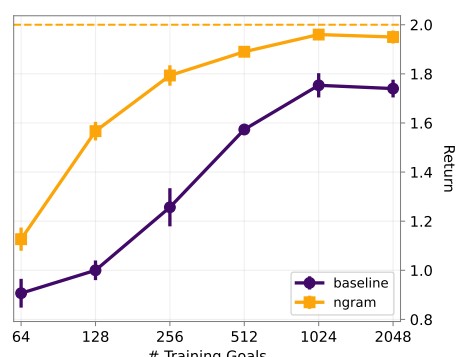

Figure 1: Performance comparison for different number of training goals between our method and Algorithm Distillation (AD), an in-context reinforcement learning method [17]. Our method demonstrates similar performance with less training goals (128 vs. 512) and in general outperforms the baseline. See Section 4 for results.

Both methods require specifically curated data, which can be demanding to obtain [22]. In addition, the in-context ability itself is transient [27] and it is difficult to predict its emergence from cross-entropy loss alone [1], making the training of such models unstable and expensive in terms of training budget. Our work takes initial steps toward addressing these challenges by introducing modifications

to the transformer's attention heads, which can accelerate training and reduce the amount of data required for in-context learning to emerge.

Induction heads have been shown to be a central mechanism that allows in-context learning in transformers [23]. Edelman et al. [6] studied the emergence of these statistical induction heads on synthetic data and concluded that transformers obtain a simplicity bias towards plain uni-grams. Akyürek et al. [2] take a step forward in this direction, demonstrating that in the in-context learning setting, the attention mechanism develops higher-order induction heads. These heads are responsible for recognizing and capturing n-grams within a sequence. Authors propose to hardcode this mechanism into a transformer, creating an n-gram layer which is used as a drop-in replacement for the multi-head attention mechanism. Intuitively, a transformer benefits from it by not learning this complicated behavior by itself; rather, it straightforwardly receives an inductive bias that n-gram heads provide. This approach significantly decreases perplexity even when applied to recurrent sequential models, indicating that n-grams play a major role in building effective in-context learning models.

In our work, we propose integrating an n-gram induction head into the ICRL model. As we demonstrate, these heads can improve model performance in low-data settings and reduce hyperparameter sensitivity while introducing only a few additional hyperparameters that are straightforward to optimize. We provide experimental evidence on Dark Room, Key-to-Door and Miniworld environments, covering both discrete and visual observation spaces.

To summarize our main contributions, in this paper we show that **N-Gram attention heads**:

- **Decrease the amount of data needed for generalization on novel tasks.** By utilizing n-gram heads, it is possible to reduce the total number of transitions in training data by a maximum of **27x** compared to the original method of Laskin et al. [17]. The results are presented in Section 4.1.

- **Help mitigate hyperparameter sensitivity in ICRL models, contributing to more stable training.** By employing n-gram heads, one may need less time searching for a good set of hyperparameters. The results are presented in Section 4.2.

- **Can be used in the environments with visual observations.** However n-grams are originally found in discrete structures (e.g. natural language texts), we show it is possible to detect repeating patterns in the sequences of images. The details of the implementation are presented in Section 2.3 and the results of the experiment are shown in Section 4.3.

## 2 METHOD

### 2.1 ALGORITHM DISTILLATION

We build our method on Algorithm Distillation [17] and use it as our baseline. It is an in-context reinforcement learning algorithm that distills the policy improvement operator by training a transformer model on specifically acquired data. As training data, the authors propose to use the learning histories of many RL algorithms that are trained to solve different tasks in the multi-task environment. After pretraining on such data, the model is able to solve unseen tasks entirely in-context by interacting with an environment without explicitly updating weights of the model.

More formally, if we assume that a dataset $\mathcal{D}$ consists of *learning histories*, then

$$\mathcal{D} := \left\{ (\tau_1^g, ..., \tau_n^g) \sim \left[ \mathcal{A}_g^{source} | g \in \mathcal{G} \right] \right\},$$

where $\tau_i^g = (o_1, a_1, r_1, ..., o_T, a_T, r_T)$ is a trajectory generated by a source algorithm from $\mathcal{A}_g^{source}$ for a goal $g$ from a set of all possible goals $\mathcal{G}$, and $o_i, a_i, r_i$ are observations, actions and rewards, respectively.

Such data might be difficult to obtain, since the aforementioned process requires training thousands of RL algorithms solving different tasks to obtain enough learning histories. In addition, AD suffers the same problems as any in-context algorithm. Learning the optimal solution can be delayed by a tendency of transformers to learn simple structures at first [6]. Moreover, the nature of in-context

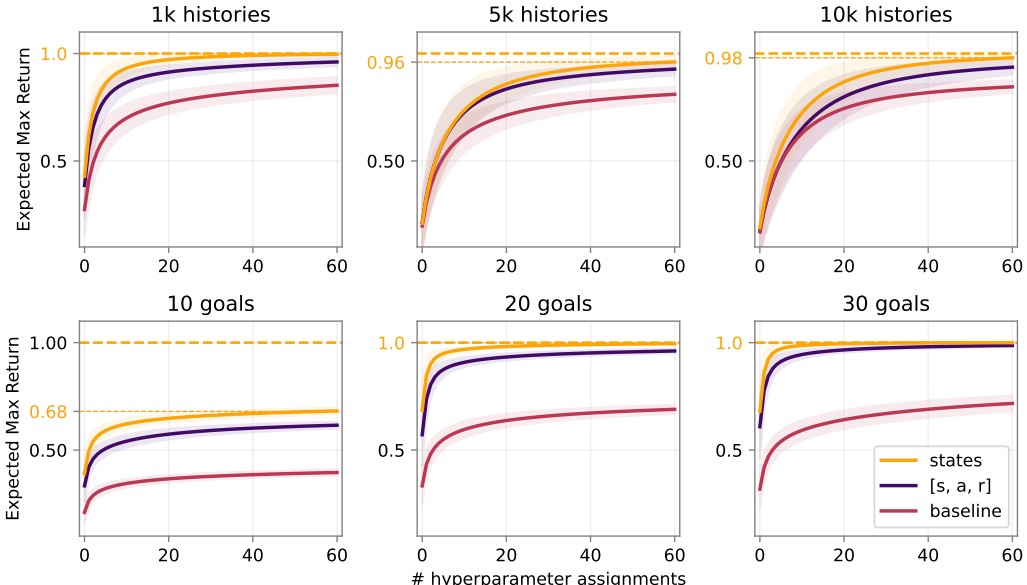

Figure 2: Results on Dark Room. We search through hyperparameters in random order and report expected maximum performance [5]. We also constrain the number of optimization steps by 10K and use equal batch size to ensure both methods use the same amount of data. **The top row** shows experiments with different number of learning histories, with the total number of training goals fixed. It is seen that our method needs much less hyperparameter assignments (20 for 1K histories) to find the optimal model, while the baseline performance increases only asymptotically (full plots are shown in Appendix D). The number of traning tasks for this experiment is 60. **The bottom row** presents experiments with varied number of goals and fixed number of learning histories. Our method makes it possible to find the optimal hyperparameters with only 15 hyperparameter assignments, while the baseline fails to work in such low data conditions. However, none of the methods can learn to generalize from only 10 goals. The number of learning histories for this task is 1K.

ability is unstable and can fade into in-weights regime as the training progresses, considerably complicating the emergence of adaptation ability [27].

## 2.2 N-GRAM ATTENTION

To address simplicity bias and improve data efficiency, we include an n-gram attention layer [2] as one of the transformer layers. This type of layer has been shown to effectively reduce simplicity bias and enhance in-context performance. Essentially, it directly incorporates the computation of n-gram statistics into the transformer, instead of relying on them to develop naturally over time. The attention pattern that is calculated from the input sequence and used in N-Gram Head (NGH) is defined as:

$$A(n)_{ij} \propto \mathbb{1}\left[(\wedge_{k=1}^{n} x_{i-k} = x_{j-k-1})\right] .$$

After that, we apply a projection and add a residual to the output:

$$\text{NGH}^n\left(h^l\right) = W_1 h^l + W_2 A(n)^\top h^l ,$$

where $n$ is the length of n-grams, $W_1$ and $W_2$ are learnable projection matrices and $h^l$ is an embedding from a previous transformer layer. In simple terms, we look for n-gram occurrences and with the help of $A(n)$ attention pattern force gradients to flow only through tokens that co-occur in the sequence.

Following Akyürek et al. [2], we also implement an N-Gram layer, which closely resembles a traditional transformer layer. The layer consists of a head $\text{NGH}^i$ that is processed through a MLP and then added to the residual stream:

$$\text{NGL}^n\left(h^l\right) = h^l + \text{MLP}[\text{NGH}^n(h^l)].$$

In the original paper, the authors used text tokens from the input sequence for n-gram matching. We lack such an opportunity when dealing with image observations, so we ought to use quantization in order to enable n-gram matching. The implementation details of the quantization process and how matching is performed are described in the Section 2.3.

## 2.3 N-Gram Matching

To find n-grams in environments with a *discrete observation* space, we use raw input sequence. However, since we are working in RL setting, the input sequence has a form of $(s_0, a_0, r_0, \ldots, s_n, a_n, r_0)$, so in our experiments we tested two approaches. We either compare the equivalence of full transitions $(a_{i-1}, r_{i-1}, s_i) = (a_{j-1}, r_{j-1}, s_j)$ or just states $(s_i = s_j)$.

In case of *pixel-based observations*, We cannot directly match raw images, as even slight variations can result in a mismatch. To address this, we use Vector Quantization (VQ) [29, 9] to quantize observations into the vectors from a codebook. We pretrain a ResNet [13] encoder-decoder model with a VQ bottleneck, which is trained to reconstruct the input image. After pretraining, each image is mapped into a $4 \times 4$ matrix of indices, and we use these for the n-gram matching. We count a match only if all the indices in the matrix are equal.

Before training starts, we use the VQ model to label images from a dataset with their indices and then train both causal and n-gram attention layers simultaneously. During the evaluation, we only make a forward pass of the VQ model in order to get the latent vectors and indices for n-gram matching.

## 3 Experiment Setup

### 3.1 Environments

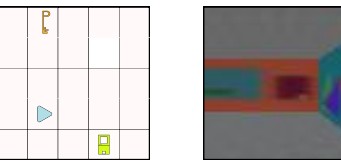

**Dark Room** is an MDP grid-world environment with discrete state and action spaces The grid size is $9 \times 9$, where an agent has 5 possible actions: up, down, left, right and do nothing. The goal is to find a target cell, the location of which is not known to the agent in advance. The episode length is fixed at 50 time steps, after which the agent is reset to the middle of the grid. The reward $r = 1$ is given for every time step the agent is on the goal grid, otherwise $r = 0$. The agent does not know the position of the goal, hence it is driven to explore the grid. The environment consists of 80 goals in total, excluding the starting square.

Figure 3: **(Left)** The Key-to-Door environment. The key and the door are shown for illustrative purposes only; the agent does not see their location during training. **(Right)** An observation from the Miniworld environment.

**Dark Key-to-Door** is a POMDP environment, similar to Dark Room, but with a more complicated task. The agent first needs to find a square with a key, and then only to find a door. The reward is given when the key is found ($r = 1$) and once the door is opened (also $r = 1$), after which the episode ends. The agent then resets to a random grid. The maximum episode length is 50, and since we can control the location of the key and door, there are around 6.5k possible tasks. The key difference of Key-to-Door compared to Dark Room is that an agent needs to use the memory to recall whether or not the key was collected to adapt its exploration strategy and successfully solve the task. We do not provide any hints after the key was collected, which makes the environment only partially observed.

Both *Key-to-Door* and *Dark Room* serve as a good starting point for testing the in-context ability in an RL setting. Despite its simple grid-structure, AD still needs a substantial amount of data to start

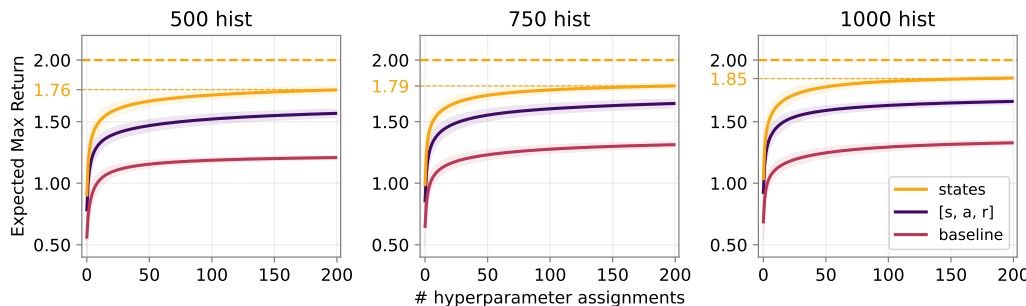

Figure 4: Results on Key-to-Door. We demonstrate the ability of our method to generalize when the task diversity is limited. We fix the total number of goals with 100, significantly shrinking the number of learning histories. Keep in mind that for the baseline method to converge to a model with the same performance, it needs 2048 goals and 2048 learning histories [17]. We show that our method needs **27x** less data comparing to baseline (see Appendix B for justification). The baseline method can no longer converge with that few data and its performance plateaus with the increasing number of hyperparameter assignments, while N-Gram model shows near-optimal performance.

showing decent performance, and these environments serve as a testbed to show N-Gram Layers help with data efficiency.

**Miniworld** is a 3D environment with an RGB $64 \times 64$ images as observations and a discrete action space. We test our method in two settings of Miniworld, the first resembling Dark Room and the latter Key-to-Door. The agent can perform three actions: move a step ahead and turn the camera left or right, no lateral movement is allowed. The episode length is 50 for Miniworld-Dark Room and 100 for Miniworld-Key-to-Door.

The Miniworld-based environments are of special interest, because while it was trivial to search for n-grams with discrete states, pixel-based observations are not so easily comparable. The details of n-grams matching for Miniworld are described in Section 2.3.

### 3.2 EVALUATION PROTOCOL

We set up and follow a specific evaluation protocol to showcase the benefits of using N-Gram layers in the ICRL setting. We use a random search over the hyperparameter space. Reporting aggregated hyperparameter search results instead of cherry-picking the best runs allows us to demonstrate the hyperparameter sensitivity of each method. To ensure that in each experiment a model has processed an equal amount of data, we fixed the batch size and limited the number of gradient steps during a run to 10K.

We evaluate the models on previously unseen goals that were not included in the training dataset. In the Dark Room environment, the number of evaluation goals varies across experiments and corresponds to all goals excluded from the training set. For instance, if a model is trained on 20 goals, it is evaluated on the remaining 60 goals. For Key-to-Door evaluation, we use 100 unseen goals and 50 unseen goals for Miniworld Key-to-Door.

To show the difference between our method and the baseline, we choose to report the Expected Maximum Performance metric (EMP) [5, 16]. By doing so, we do not report the best performance of a single checkpoint, rather we show the expected maximum performance for a certain computational budget. Using this approach, we simultaneously compare our method with a baseline in terms of ease of training and maximum achieved performance. The exact hyperparameter assignment setups are shown in Appendix C.

### 3.3 DATA COLLECTION

Algorithm Distillation introduce several requirements on the structure of the data. It should be comprised of learning histories, i.e. there should be an implicit ordering in data from the least to the most effective policy. To produce such histories, we used a combination of approaches.

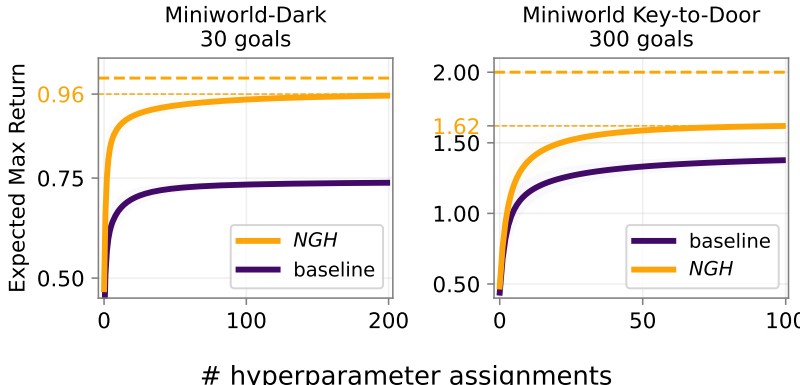

Figure 5: Results on Miniword environments. We show that our method is applicable not only for environments with discrete observations, but also for the image-based ones. The settings of the Miniworld environments are similar to Dark Room and Key-to-Door. The main outcome of these experiments is that we can successfully implement n-gram matching in for images and get similar results to the discrete environments. The details of the setup are described in Section 4.3. **(Left)** We fix the number of tasks in Miniworld-Dark to 30 and the number of learning histories to 50. The N-Gram layer significantly enhances performance, resulting in nearly-optimal model, while the baseline quickly saturates around suboptimal return. The evaluation is done on 50 goals. **(Right)** For the experiment in Miniworld-Key-to-Door we fix 300 training tasks and 50 learning histories. The results follow the pattern observed in discrete environments, where the N-Gram layer refines the performance of the baseline model. We evaluate the models on 100 unseen goals.

For grid-world environments, we use a table Q-Learning algorithm [31] and save $(s_i, a_i, r_i)$ transitions. In image-based environments, we use the approach described in Zisman et al. [33]. For this, we implement an oracle agent and design a decaying noise schedule. It allows us to collect the learning histories faster than training any model-free RL algorithm from scratch for each task. The rest of the data collection process remains unchanged.

Throughout the text we use the terms *learning histories* and *tasks*. The task is a predefined grid or a pair of grids an agent must come to upon it receives a reward. The learning history is an ordered collection of states, actions and rewards an RL algorithm observed (or produced) while learning to solve a *single* task. When we say we generated a dataset of $n$ tasks with $m$ learning histories, it means for each of the task there are at least $\lfloor \frac{m}{n} \rfloor$ learning histories per task. Unlike Laskin et al. [17], we distinguish between tasks and learning histories, as it is often the case with real data when many trajectories correspond to only a few tasks [32, 8].

## 4 RESULTS

In this section, we examine how ICRL models can benefit from N-Gram layers and explore potential challenges associated with their use. We analyze their role in hyperparameter search efficiency, data efficiency, and applicability to image-based observations. Additionally, we examine whether they significantly expand the hyperparameter search space or negatively affect baseline performance.

### 4.1 N-GRAM LAYERS CAN MAKE THE SEARCH FOR OPTIMAL HYPERPARAMETERS QUICKER

In-context learning is known for its instabilities: it is difficult to predict an emergence of in-context ability from the loss function value [1]; it is transient, meaning that during training it can switch between in-weight and in-context regimes [27]. Because of these drawbacks, finding a good set of hyperparameters can be sufficiently delayed, which leads to more resources being spent on computation. We hypothesize that by including n-gram heads from the start, rather than waiting for their emergence during training, we can adequately decrease the computational budget and make the hyperparameter search faster.

To demonstrate the effect of N-Gram layers on the hyperparameter sensitivity of the model, we perform a random search over the core transformer hyperparameters that do not change the parameter count of the model. The effect of N-Gram heads is illustrated in Figure 2. In the top row, we fix the number of training tasks at 60 and vary the number of learning histories. It can be seen that the model with n-gram layers can find the optimal parameters faster than the baseline model. For 1K learning histories, finding the optimal model requires just over 20 hyperparameter assignments, while the baseline model needs more than 400. When the number of tasks varies, the baseline model quickly saturates at suboptimal performance and asymptotically improves thereafter, whereas the n-gram model reaches optimal performance in about 15 assignments. Full-length plots are available in Figure 9.

## 4.2 N-Gram layers improve data-efficiency of ICRL algorithm

In-context reinforcement learning imposes special limitations on data, making them difficult to obtain. Moreover, the performance of the ICRL model can be affected by meta-parameters of the data [33], such as the diversity of tasks, the number of learning histories per task, and the learning pace of data-generating RL algorithm.

In real-world data, there are often many trajectories per task, but the number of distinct tasks is limited. [32, 8]. In such cases, a desirable quality of the model is its ability to avoid overfitting on the training data while generalizing to unseen tasks. Our hypothesis here is that incorporating N-Gram layers into the model can help build a more data-efficient model and enhance generalization by capturing sequential patterns within trajectories.

To show the effect of N-Gram layers when task diversity in data is low, we set up an experiment in the Key-to-Door environment, since it possesses 6.5K tasks in total. To simulate low task diversity, we fix the number of training goals by 100 and sample another 100 unseen goals for evaluation. It can be observed from Figure 4 that the baseline method is struggling to produce a model that is able to generalize to unseen goals in such a low data setting. In turn, our method demonstrates performance on par with what Laskin et al. [17] report in their work. We note that compared to AD, our method needs 27x less data, detailed computations are provided in Appendix B.

## 4.3 N-Gram layers can be used with images as observations

It is relatively straightforward to match n-grams in discrete settings, like text or grid-world environments. The problem arises when the observation space is image-based. We cannot directly compare the images, as even a slight camera rotation would invalidate a match; however, they may still correspond to the same state.

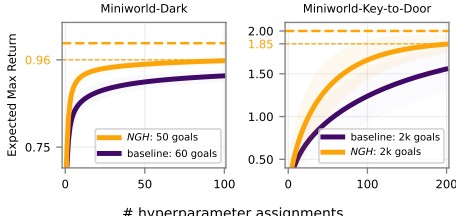

To address this, we need a model that disregards minor differences in its encoding and instead focuses on state-representative details, such as the color of the wall the agent sees and its distance from the wall. We utilize the Vector Quantization [29] technique for this reason, the details of n-gram matching are described in Section 2.3.

Figure 6: Hyperparameter sensitivity. **(Left)** Results on Miniworld-Dark. The N-Gram layer model is trained on 50 goals, the baseline model is on 60. For evaluation, 20 goals were used. **(Right)** Results on Miniworld-Key-to-Door. Both N-Gram and baseline models were trained on 2K goals and evaluated on 100 unseen goals.

We transfer the Dark Room and Key-to-Door setting into a 3D environment Minigrid, where an agent receives a $3 \times 64 \times 64$ RGB image as an observation. We observed similar differences in performance of the N-Gram and baseline models. N-Gram layer is able to reduce the number of hyperparameter assignments needed to find a model with near-optimal performance in both Miniworld-Dark (Room, omitted for brevity) and Miniworld-Key-to-Door environments, see Figure 6. In a low-data regime, N-Gram layers also improve performance compared to the baseline. As shown in Figure 5, N-Gram layers enhance performance in both environments.

## 4.4 N-GRAM LAYERS DO NOT SIGNIFICANTLY EXPAND HYPERPARAMETER SEARCH SPACE

N-Gram layer introduce new hyperparameters to optimize, such as n-gram length and position of the layer to which N-Gram layer is inserted. A natural question arises: do these hyperparameters also require extensive search, and how sensitive is the model to them?

To address this question, we conducted six random hyperparameter searches in Miniworld-Dark, ablating either the layer position or the n-gram length while keeping one variable fixed. For the n-gram length search, we fixed the position at [1] (after the first layer), whereas for the layer position HP search, we set the n-gram length to 1. Following [2], we do not insert N-Gram layer as the first or last layer. While searching for the optimal n-gram length, we consider "up to" a given n-gram. For example, a 2-gram includes both a 1-gram and a 2-gram together. We continue to report the EMP metric, but here we present only the final value achieved after all hyperparameter assignments (full plots are available in Appendix D).

Table 1(a) and Table 1(b) show that there is no significant difference between neither the n-gram length, nor the position of the N-Gram layer inside a transformer. This may indicate that there is little to no overhead in hyperparameter search caused by introduction of N-Gram layers.

## 4.5 INSERTING N-GRAM LAYERS DOES NOT HURT THE PERFORMANCE OF A BASELINE ALGORITHM

Another concern when working with N-Gram layers is whether they can affect the performance of a baseline model. Hypothetically, this can occur if the quantization model fails to correctly identify which image observations correspond to the same underlying state, rendering the n-gram matching mechanism ineffective.

We designed the following experiment to test this hypothesis. Using VQ as an n-gram extraction tool, we follow the standard procedure described in Section 2.3, with one key modification. After matching, we shuffle the n-gram attention matrix $A(n)_{ij}$, effectively simulating a completely ineffective N-Gram attention layer that selects incorrect observations as n-gram matches. Like in the previous experiment, we run a random HP search in Miniworld-Dark environment and report the EMP calculated for the last hyperparameter assigned.

We compare the model with the permuted n-gram mask with the baseline model without the N-Gram layer, the results are shown in Table 1(c). No significant difference is observed between the two models, suggesting that when the n-gram matching mechanism is flawed, the model's performance remains comparable to that of a model without an N-Gram layer.

## 5 RELATED WORK

**In-context RL.** The key feature behind ICRL is the adaptation ability of a pretrained agent [18, 28]. In general, it relies on the transformer's ability to infer a task from the history of interactions with an environment. Müller et al. [20] show that transformers are capable of Bayesian inference, which is known for its applicability to reasoning under uncertainty [10]. Laskin et al. [17] proposed to pretrain a transformer on the learning histories of RL algorithms which allows it to implicitly learn the policy improvement operator. During inference on unseen tasks, a transformer is able to improve its policy by observing a context and inferring a task from it. However, such an approach requires specific datasets, which may be expensive to collect [22]. To address this, it has been proposed to generate datasets following the noise curriculum instead of training thousands of RL agents [33], perform augmentations of existing data [14] or filter out irrelevant data [26]. Our work follows the direction of loosening data restrictions, but instead of working with data, we introduce a model-centric approach, making a transformer to demonstrate in-context abilities while operating on a restricted amount of data.

**N-Gram and Transformers.** N-Gram statistical models have been known for decades and used in the statistical approach to language modeling [3, 15]. More recent approaches [25, 19] study the application of n-grams to transformer models, finding that they can increase overall performance. Akyürek et al. [2] discover that a transformer implicitly implements the 2-gram attention pattern when solving the in-context learning task, which authors denote as a higher order of induction head [23].

They explicitly implement 1-, 2-, and 3-gram attention layers and observe a significant reduction in perplexity of the pretrained models. Another work [6] directly investigates the behavior of n-gram induction heads during the training process. The authors find that transformers are biased towards simple solutions, thus making it problematic for higher-order induction heads to appear. To our knowledge, we are the first to apply these findings in a decision-making setting.

## 6 CONCLUSION AND FUTURE WORK

In our work we show that incorporating n-gram induction heads can sufficiently ease training of in-context reinforcement learning algorithms. Our findings are threefold: (**i**) we show that n-gram heads can fairly decrease sensitivity to hyperparameters of ICRL models; (**ii**) we demonstrate that our method is able to generalize from much fewer data than the baseline Algorithm Distillation [17] approach. (**iii**) however the original n-gram heads were designed for discrete spaces, we showed it is possible to adapt the approach to environments with visual observations by utilizing vector quantization techniques. We speculate that n-gram heads are useful in ICRL due to the imperfect nature of in-context learning itself: a tendency of transformers to converge to simple solutions first [6], and the transitivity of the in-context ability itself [27].

Although we believe our findings are promising, there are some limitations of the current work. Further research is needed to investigate the behavior of N-Gram heads in more comprehensive environments, e.g. XLand-Minigrid [21] or Meta-World [32]. Additionally, while image observations account for a significant portion of RL applications, exploring methods to apply N-Gram heads to proprioceptive continuous states could provide further insights.

Table 1:

(a) Ablation on n-gram length

| N-Gram max | EMP |
|---|---|
| 1-gram | $0.74 \pm 0.02$ |
| 2-gram | $0.71 \pm 0.01$ |
| 3-gram | $0.76 \pm 0.05$ |

(b) Ablation on N-Gram layer position

| Position | EMP |
|---|---|
| [1] | $0.69 \pm 0.03$ |
| [2] | $0.69 \pm 0.02$ |
| [1, 2] | $0.67 \pm 0.005$ |

(c) Comparison of baseline and a random n-gram mask

| Model | EMP |
|---|---|
| Permuted | $0.51 \pm 0.03$ |
| Baseline | $0.52 \pm 0.02$ |

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

## A  APPENDIX

## B  CALCULATION OF TRANSITIONS IN DATA

In appendix I of Laskin et al. [17] they mention that AD is more data-effective than source algorithm and report the size of a dataset. The total number of data needed to achieve an approximate of 1.81 return on Key-to-Door [1] is reported as

> (...) on 2048 Dark Key-to-Door tasks for 2000 episodes each.

The estimate of total number of transitions *to generate* for AD, considering the maximum length of an episode in Key-to-Door is 50 steps, equals: $2048 \times 2000 \times 50 = 204.8\text{M}$ transitions.

We generate 100 unique training tasks and then sample 750 train task with repetition from the original 100. Then we make 200 training episodes for each task. In total, we get $750 \times 200 \times 50 = 7.5\text{M}$ transitions, which is more than **27x** less data.

## C  HP SEARCH SETUPS

We use weights and biases sweep for running sweeps. All of the sweep setups are available by this clickable link [will be available after de-anonymization].

We also report the setup of hyperparameter sweep in the table below.

The experiments are run on H100 cluster, the experiments took around 40K GPU-hours throughout the project, including failed runs.

---

[1] since no accurate data of plots was published, we used free-to-use WebPlotDigitizer for Fig. 6 in AD paper

Table 2: Hyperparameter Configurations

(a) Grid Environments

| Parameter | Distribution | Values |
|---|---|---|
| batch size | - | 1024 |
| embedding dropout | Uniform | [0.0, 0.9] |
| seq len | - | [60, 100, 160, 200] |
| subsample | - | [4, 8, 10, 20, 50] |
| residual dropout | Uniform | [0.0, 0.5] |
| ngram head pos | - | [1], [2], [1, 2] |
| ngram max | - | [1, 2] |
| label smoothing | Uniform | [0.0, 0.8] |
| learning rate | Log Uniform | [1e-4, 1e-2] |
| weight decay | Log Uniform | [1e-7, 2e-2] |
| pre norm | - | [true, false] |
| normalize qk | - | [true, false] |
| hidden dim | - | 512 |
| update steps | - | 10000 |

(b) MiniWorld environments

| Parameter | Distribution | Values |
|---|---|---|
| batch size | - | 1024 |
| embedding dropout | Uniform | [0.0, 0.8] |
| seq len | - | [100, 150, 200] |
| subsample | - | [8, 16, 32] |
| residual dropout | Uniform | [0.0, 0.8] |
| ngram head pos | - | [1], [2], [1, 2] |
| ngram max | - | [1, 2] |
| label smoothing | Uniform | [0.0, 0.8] |
| learning rate | Log Uniform | [5e-4, 1e-2] |
| weight decay | Log Uniform | [1e-7, 2e-2] |
| pre norm | - | [true, false] |
| normalize qk | - | [true, false] |
| hidden dim | - | 512 |
| update steps | - | 10000 |

## D    FULL PLOTS

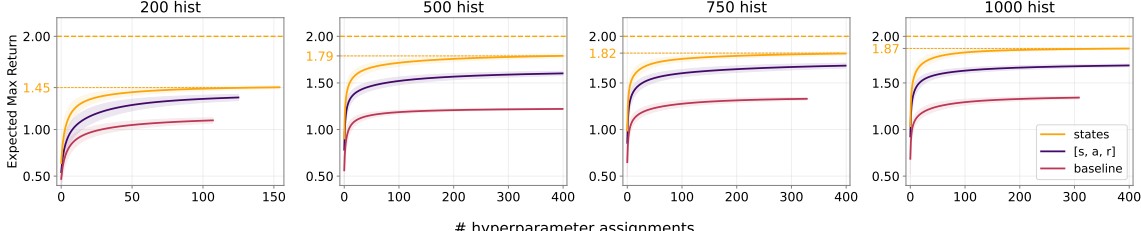

Figure 7: Full length plots for Key-to-Door. For 200 learning histories we halted the random search early, since it was obvious the performance has stalled.

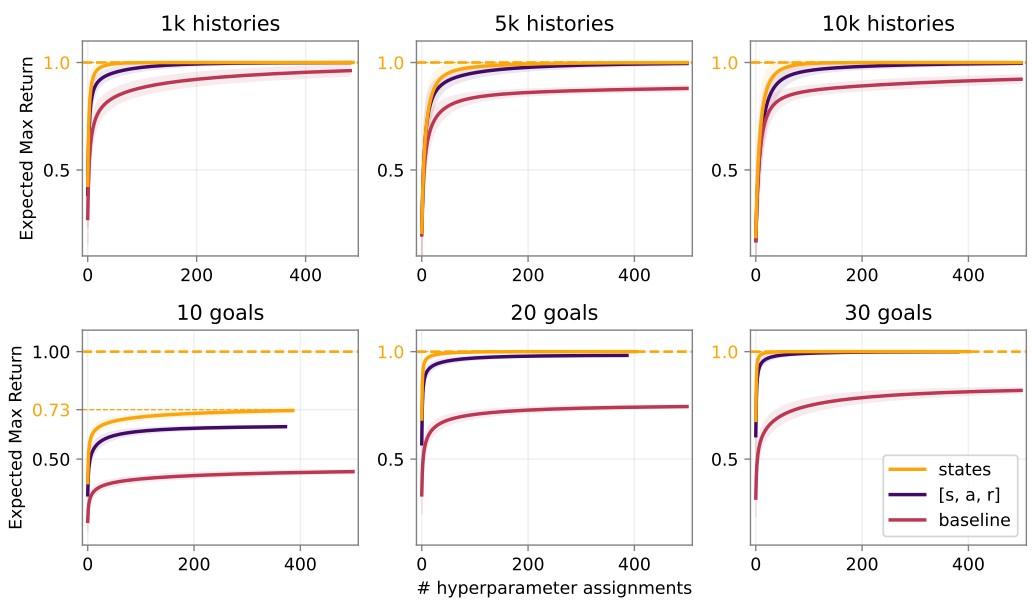

Figure 8: Full length plots for Dark Room. Some of the computations halted earlier for the same reason as in Figure 7

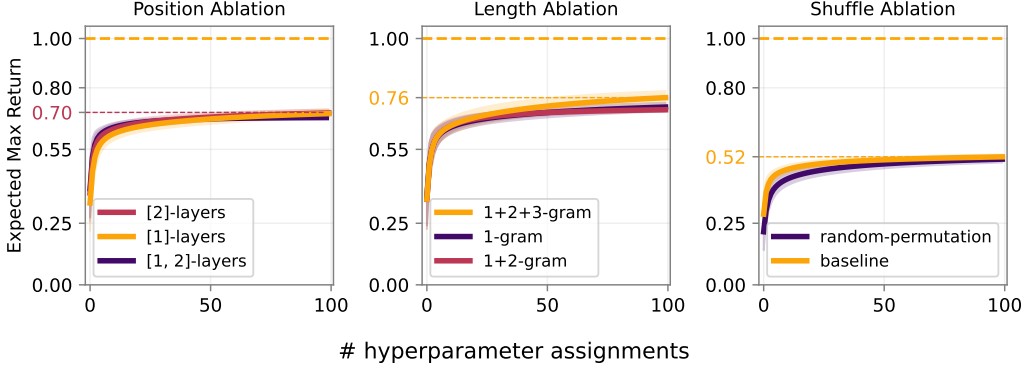

Figure 9: Full length plots for ablation experiments in Miniworld-Dark environment.

# E   PERFORMANCE OF AD ON KEY-TO-DOOR AND DARK ROOM

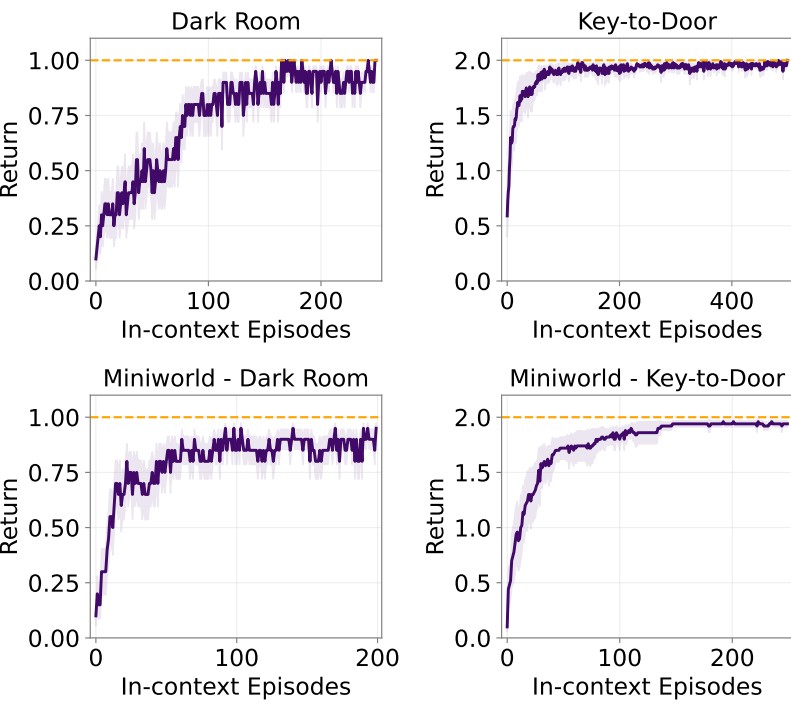

Figure 10: AD performance on Dark Room and Key-to-Door. This plot shows that our implementation of AD demonstrates optimal performance given the right hyperparameters.

