# OpenReview forum: "N-Gram Induction Heads for In-Context RL: Improving Stability and Reducing Data Needs"
_ICLR.cc/2026/Conference — Submitted to ICLR 2026_

### Official Review · Reviewer_yhM1 · 2025-10-24

**Soundness:** 2
**Presentation:** 2
**Contribution:** 2
**Rating:** 2
**Confidence:** 4

**Summary:**

This paper proposes to explicitly add n-gram induction heads to transformer model for in-context reinforcement learning. It can improve the data efficiency, address the simplicity bias and make the model more stable and less sensitive to hyperparameters. Experiments show that the proposed method can outperform the algorithm distillation baseline.

**Strengths:**

1. The idea of introducing n-gram induction head is simple and easy to implement.
2. It can improve the data efficiency and reduce the hyperparameter sensitivity.
3. The proposed n-gram head can also be applicable to visual environment settings.

**Weaknesses:**

1. Section 2.2: the description of the whole model structure is unclear. The formula in line152 needs more detailed explanation, like how the attention score of n-gram head is normalized, what if there are no n-grams in the history, if there is only one head in each NG layer. The authors should better provide a pseudo-code or the architecture figure.

2. What is the architecture of your base transformer model? What is the number of layers and the total size of the model? Do you try using a powerful pretrained model to see if the benefit of 2-gram information still helps? There should also be experiments on different model size to further validate the effectiveness of the n-gram head.

3. The limited generalizability. The motivation of the 2-gram induction head  is to mitigate the simplicity bias and accelerate the emergence of in-context learning.
While effective at small scales and with limited data, this architectural improvement is fundamentally a hardcoded bias for simple, short-term pattern recognition (specifically, 2-gram statistics in the experiments, 3-gram will hurt the performance). In the era of increasingly large transformers and datasets, the attention itself can emerge the complex induction heads and even more sophisticated learning algorithms, which makes the 2-gram information redundant. Moreover, in real-world scenarios, the basic in-context learning capability is already strong, we do not need to train the transformer from scratch for the emergence of in-context RL, and the n-gram information is insufficient for distilling complex RL algorithms that require abstract reasoning instead of the local sequential patterns.

**Questions:**

Typos: The title of the paper is wrong.

---

### Official Review · Reviewer_vnMf · 2025-10-27

**Soundness:** 2
**Presentation:** 2
**Contribution:** 2
**Rating:** 4
**Confidence:** 2

**Summary:**

This paper proposes N-gram attention in ICRL to reduce the data required in training process but achieve the similar performance of AD. Evaluations on some visual-based scenarios further show its edge.

**Strengths:**

1. The paper is easy to follow.
2. The figure is well-situated and clear.

**Weaknesses:**

As I'm not the expert in this area, I cannot provide some professional review here. Below are my comments:

1. The work lacks formal analysis explaining why N-gram heads yield stability gains and reduced data requirement. Maybe the authors should provide some theoretical perspective to elaborate this.
2. Could the authors add more experiments on some diverse visual tasks to further showcase the superiority ?

**Questions:**

See weakness above

**Details Of Ethics Concerns:**

Format issue: The title in pdf is not corresponding with the title on openreview.

---

### Official Review · Reviewer_ULwF · 2025-10-29

**Soundness:** 2
**Presentation:** 3
**Contribution:** 2
**Rating:** 6
**Confidence:** 3

**Summary:**

This paper integrates n-gram introduction heads into transformers for In-Context RL. This integration aims to resolve two challenges in Algorithm Distillation: (1) the low data efficiency in ICRL; (2) the training instability in emerge the in-context capability. The authors apply explicit n-gram introduction heads to the transformer backbone and test n-gram heads in Dark Room and Key-to-Door environments. The authors also propose a scheme to extend the environments to image states. The results show that the n-gram heads can significantly reduce the data requirement and improves hyperparameter robustness.

**Strengths:**

1. This paper proposes a very useful improvements to AD by integrating n-gram heads to transformer. This architectural improvement significantly mitigates the disadvantage of AD of data overhead, hyperparameter tuning, and training instability
1. The empirical study is very comprehensive that answers all my concerns towards the n-gram heads architecture application in AD. Section 4 strengths my confidence in applying this technique to ICRL research.
1. The VQ technique provides a feasible approach to image-based environments which extends the n-gram heads architecture to broader application scenarios.
1. The authors give detailed description on how to sweep and choose the hyperparameters. This information is rather important to such empirical papers for reproduction.

**Weaknesses:**

1. This paper demonstrates the benifits of n-gram heads by expriments in two environments and their variants in Miniworld. However, these environments are relatively simple. I suggest the authors test n-gram heads in more complicated continuous control tasks (e.g., MuJoCo, Meta world) to show the generalization.
1. In addition to the experiments, I would expect theoretical analysis for why explicit n-gram heads can help improve data efficiency and traning stability in Algorithm Distillation, as n-gram has been studied for decades.
1. As VQ and ResNet are used in image-based tasks, I suggest the authors also analyze the training cost sensitivity for these modules. They are important to prove the adaptation and efficiency in pixel-based environment.

Minor issue:
1. The title is missing in the paper.
1. The figure positions mismatch the text. For example, I have to jump pages to find Fig. 2 and Fig. 4 when I read the paper. It would be nice to rearrange the paper so that the figures are right in the place where they are referred.

**Questions:**

1. Could the authors compare with other baselines, for example, Noise Distillation [1] mentioned in the paper? This will help clarify whether the observed data-efficiency and stability gains stem specifically from the proposed n-gram inductive bias, or whether similar improvements could be achieved through alternative data-centric approaches.
1. Could the authors test the method in high dimensional continuous control environments, e.g., MuJoCo and Meta world? These are widely used benchmarks to evaluate the generalization and scalability of RL algorithm, and would help demonstrate whether the proposed n-gram heads can extend the scope to more complex, real-world control tasks.

[1] Zisman, Ilya, et al. "Emergence of In-Context Reinforcement Learning from Noise Distillation." International Conference on Machine Learning. PMLR, 2024.

---

### Official Review · Reviewer_vyuv · 2025-10-29

**Soundness:** 2
**Presentation:** 1
**Contribution:** 1
**Rating:** 2
**Confidence:** 3

**Summary:**

This paper examines whether augmenting the architecture with n-gram attention heads improves the performance of Algorithm Distillation [17].

**Strengths:**

N/A

**Weaknesses:**

The scope of this work is narrow, as it focuses solely on improving Algorithm Distillation [17], and it is unclear whether the findings generalize beyond this specific approach. Given the limited experimental scale, its relevance to understanding in-context learning in large transformer models is also unclear.

**Questions:**

The title in the PDF file ("FORMATTING INSTRUCTIONS FOR ICLR 2026 CONFERENCE SUBMISSIONS") should be corrected.

---

### Official Review · Reviewer_Csny · 2025-10-31

**Soundness:** 2
**Presentation:** 2
**Contribution:** 1
**Rating:** 2
**Confidence:** 4

**Summary:**

This paper integrates n-gram induction heads into transformer architectures for in-context reinforcement learning (ICRL). Building on Algorithm Distillation, the authors show that explicitly incorporating n-gram attention patterns reduces data requirements (up to 27× fewer transitions) and makes hyperparameter search easier, and works across both discrete and pixel-based environments. The method is validated on Dark Room, Key-to-Door, and Miniworld environments.

**Strengths:**

S1: The paper addresses a real practical problem in ICRL: Algorithm Distillation's demand for large datsets, and sometime slow convergence or getting stuck.

S1: The paper tests their method on different setups, ones with more grid based states and another with rgb 64x64 pixel. A diverse set of tasks reinforces the findings.

S3: Clear presentation with intuitive figures.

**Weaknesses:**

W1: This is my main concern. This work seems to not disentangle three questions: 1) how does n-gram heads general help the training of transformers 2) why this is espeically important (or not) in ICRL tasks 3) is the gain via n-gram heads really not achievable by other hyperparameter tuning or different architectural tricks.

W2: It is very hard to takeaway generalizable insights on why n-gram heads help and when we should be using these.

W3: Lack of serious comparison. The method is only compared against vanilla AD. There has been many variants on the algorithmic side, and at the same time many improvements on the transformer architecture and initialization.

W4: Overselling of generality. The title and abstract suggest n-grams solve ICRL's data problems broadly, but it is hard to understand whether these gains are precisely attributable to n-gram heads or will be simply addressed by data scale or better optimization.

**Questions:**

Q: What is the positional encoding used in the transformer?

---

### Meta-Review · Area_Chair_EW6x · 2026-01-02

**Summary:**

The reviewers had concerns that the paper was not thorough enough: a number raised issues with the lack of details on the implementation; others had issues with the lack of diverse datasets on which the method's purported benefits could be verified; others had issues with a lack of theoretical understanding of the method proposed.  I think one of these drawbacks by itself would have been fine, but with all three issues present to a relatively high degree, I cannot recommend acceptance.  I recommend that the authors either invest effort in analyzing their method in a larger suite of RL tasks, or by working towards a better theoretical understanding of the method.

**Reviewer Concerns:**

The authors did not give a rebuttal so none of their concerns were addressed.

**Reviewer Scores:**

Since there was no rebuttal, I can assume that all scores would be maintained or reduced.

---

### Decision · Program_Chairs · 2026-01-26

Reject